# Analysis of Cooperative Perception in Ant Traffic and Its Effects on Transportation System by Using a Congestion-Free Ant-Trail Model

**DOI:** 10.3390/s21072393

**Published:** 2021-03-30

**Authors:** Prafull Kasture, Hidekazu Nishimura

**Affiliations:** Graduate School of System Design and Management, Keio University, Yokohama, Kanagawa 223-8526, Japan; h.nishimura@sdm.keio.ac.jp

**Keywords:** cooperative perception, congestion-free transportation, ant traffic, communication, intelligent transportation systems, decentralize transportation, self-organisation in transportation

## Abstract

We investigated agent-based model simulations that mimic an ant transportation system to analyze the cooperative perception and communication in the system. On a trail, ants use cooperative perception through chemotaxis to maintain a constant average velocity irrespective of their density, thereby avoiding traffic jams. Using model simulations and approximate mathematical representations, we analyzed various aspects of the communication system and their effects on cooperative perception in ant traffic. Based on the analysis, insights about the cooperative perception of ants which facilitate decentralized self-organization is presented. We also present values of communication-parameters in ant traffic, where the system conveys traffic conditions to individual ants, which ants use to self-organize and avoid traffic-jams. The mathematical analysis also verifies our findings and provides a better understanding of various model parameters leading to model improvements.

## 1. Introduction

With the rise in communication and sensing technology, cooperative perception for intelligent transportation systems (ITS) is attracting attentions of various researchers in the field [1,2,3,4,5,6]. Cooperative perception allows vehicles to collect and share information with other vehicles and infrastructure, enabling vehicles and infrastructure to detect beyond their local capabilities [1,2]. Research has shown that cooperative perception can increase autonomous driving systems’ robustness by increasing the perception’s accuracy and detecting objects beyond their local capabilities [7,8,9]. Moreover, the research has also shown that cooperative perception can allow individual vehicles in ITS to collaborate on a transportation system level to increase the ITS system’s efficiency and safety [3,4,5,7,8,9,10].

However, although the cooperative perception might be new in vehicular transportation systems, in nature, transportation systems that use cooperative perceptions are common phenomena [11]. In the recent past, studies of cooperative perception among ants are attracting various multidisciplinary researchers. Many practically blind ant species create chemical trails to share and collect information through chemotaxis. Ants use the collected information for efficient transportation, food exploration, immigration, and the colony’s defense [12]. The transportations in ant colony have remarkable similarities with vehicular traffic systems [12,13,14,15,16]. Thus, it is argued that the collective movement of ants on trails [here onwards called as “ant traffic” (AT)] is analogous to vehicular transportation network [11,17]. In AT, considering visual sensing limitations, cooperative perception through chemotaxis (chemical communication) plays a crucial role in managing the transportation activity [12,18]. Thus, it is believed that learning about the cooperative perception and communication (CP&C) system in AT will help us to design and manage cooperative perception in ITS. In the case of AT, multiple previous studies have indicated that biological evolution may have optimized communications in AT, leading to efficient transportation systems; examples include (i) use of chemotaxis for selecting the shortest path to a food source and (ii) the formation of three lanes in bidirectional AT [12,18,19,20]. Recent experiments about unidirectional AT from a transportation perspective revealed other exciting properties of AT [21]. The studies show that ants collaboratively achieve constant average velocity independent of density through CP&C. Consequently, no jamming phase was observed in the fundamental diagrams for AT. Predominantly tendency to form clusters was also observed. The fundamental diagrams for AT are contradictory to the fundamental diagrams for vehicular traffic. A decreasing average velocity with density is observed in the latter, leading to congestions [21,22].

Motivated by AT’s new findings, several previous studies makes an effort to explain the CP&C behind congestion-free AT [13,22,23,24,25]. These studies proposed models of AT that attempt to replicate AT behaviors through computer simulation. Although the studies in [22,23,24] are successful in presenting models of AT, which can replicate the behavior of AT from [21], all of these models contradict multiple previous studies about ant physiology. Whereas in contrast, the [13]’s model is simple yet non-contradictory to previous studies [25]. Simultaneously, the model is well analyzed [11,15,16,26,27,28]. However, the model in [13] is a generic model, proposed and analyzed before [21]. Hence, it needed to be improved and analyzed while considering [21]’s findings. Therefore in [25,29,30], we presented an improved model of AT with a new perspective on the mechanism of congestion-free AT in light of new findings from [21]. Similar to [23,24], the model in [25,29,30] also assumes that ants use CP&C through chemotaxis for traffic management on AT. However, in contrast to [23,24], the studies argue that for an increase in pheromone concentration, ants increase their velocity. The modelling assumptions in [25,29,30] use references from physiological studies of ants as the base. Simultaneously the model simulation captures [21]’s main findings. Thus proving that the model represents real AT.

In the study reported in [25,29,30], an introduction and analysis of an agent-based model of AT [ant-trail model (ATM)] are presented. As shown in [30], ATM’s design and assumptions are based on an extensive survey of previous studies of ant’s physiology. The study in [30] also presents traceability between the previous studies of ant’s physiology and the ATM’s dynamics and assumption. The traceability provides us with confidence that the model dynamics and assumption in ATM represent real-life ant. On the other hand, [25,29] presents a comparison between ATM model simulations and all the empirical findings from [21,22]. As shown in [25,29], ATM simulation captures all the characteristics of real-life AT. Thus based on the finding in [25,29,30], we argue that ATM simulations not only represents real-life ant’s physiology, but it also represents real-life AT as transportation system. Hence, we argue that ATM simulations can be used for analyzing real-life AT.

The study in [25,29] also presents an analysis of AT from the traffic flow perspective. The analyses in [25,29] indicate that ants use CP&C through chemotaxis to implement a jam-absorption mechanism (JAM), which allows ants to maintain traffic flow on the trail. The studies indicate the use of a slow-in-fast-out strategy by ants for creating multiple platoons with considerable distance between the platoons (inter-platoon distance), which facilitate AT in implementing the JAM explained in [31,32]. The inter-platoon distances lead to sufficient time headway, allowing ants to absorb the queuing effect due to the platoon ahead [25,29,31,32]. Because of the jam absorption mechanism, no platoon gets caught up in the stop-and-go motion that existed in the platoon ahead, causing emergent congestion-free traffic [25,29].

However, although the studies in [25,29,30] explain JAM in AT from the traffic flow perspective, it does not elaborate on how ants use CP&C to achieve it. The studies explain the actions of individual ants in the implementation of collaborative JAM. The studies also explain how individual ants’ actions collectively lead to management of the traffic flow in AT. Yet, the studies fail to elaborate on information that ants use through CP&C to make those decisions or the communication mechanism that enables individual ants to receive that information.

To implement complex collaborative behavior such as JAM, ants require decentralized collaboration between multiple, independently operated ants. Such collaborations need to be based on a collective understanding of a larger transportation scenario than the immediate surrounding [33]. On the other hand, for the same individual ants, the decisions also need to accommodate the information about the immediate surroundings for safety and efficiency [33,34]. For the CP&C system in AT, an ideal system should facilitate the system-wide understanding of transportation scenario for collaboration while conveying the immediate surroundings for avoiding collisions [33,34]. Thus, to collaboratively implement the decentralized JAM, the individual ants need to receive information about the larger and ever-evolving traffic system while also acknowledging local surroundings. In addition, from an ITS research perspective, to design AT inspired CP&C for a congestion-free ITS, we must understand (i) the information that allows ants to understand the holistic traffic scenario and (ii) the communication system that provides that information. It is also critical to understand the environmental or communication conditions in which the communication system is functional.

Therefore, taking the lead from the introduction, validation and analysis of ATM in [25,29,30], this paper investigates various CP&C related parameters and their effects on AT. The study analyses the traffic information that individual ants receive through chemotaxis and the various ways ants use it to collaborate. We also analyzed the AT conditions and environment for the communication system in which the system conveys the information about traffic condition to individual ants. Model simulations and approximate but simple mathematical expressions of key transportation scenarios in ATM were used for the analysis. The model simulations allow us to analyze system-level emergent cooperative perception in AT. Whereas the mathematical expressions help us validate our observations and better understand the model parameters leading to model improvements. Understanding the above parameters and scenarios should clarify the critical cooperative perception behind jam absorption mechanisms in AT, which can be used to design efficient ITS.

## 2. Model and Simulation Scenario

For the readers’ convenience, we briefly re-introduce ATM from [25,29,30] here. Individual ants in AT cooperate to understand traffic scenario through CP&C. In AT, individual ants’ travelling decisions depend on traffic perception through the chemical substances known as pheromones [18,19]. As ants move forward on the trail, they excrete pheromones, which other ants can sense and follow [12,18,19,20]. The forward movements of ants depend on the local pheromone concentration (σ) ahead of them, where the detected concentration of pheromone is converted into a self-propelling force. Ant’s self-propelling force increases with increasing concentration until the concentration ($\sigma_{sat}$) that saturates the antennae [12,18,19,20,35,36]. The antennae cannot differentiate between pheromone concentrations above $\sigma_{sat}$. Thus the propulsion force for the corresponding pheromone remains approximately similar to the saturation one [12,35,36]. Based on the aforementioned chemotaxis behavior of ants, [25,29] present a model of the AT in ATM. Although the previous study in [13] uses cellular automaton for its analysis, considering that ants parameters are important for our analysis, we use agent-based modelling for our study. The agent-based ATM simulations have two types of agent, namely (i) stationary agents representing the cells of the trail (environment in the model) and (ii) moving agents representing the ants. Each cell of our one-dimensional ant trail can accommodate, at most, one ant at any time step (see Figure 1). The cells are labeled by the index i (i = 1,2,…, L), where L is the length of the trail. For the analysis, we associate the following two numerical variables with each cell.

$s_{i}\left(t\right)$ is a binary variable, which either can be 0 or 1 depending on whether the cell is empty (0) or occupied (1) by an ant at time step t.$\sigma_{i}$(*t*) is a numerical variable, which represents the pheromone concentration in the given cell. $\sigma_{i}\left(t\right)$ ranges from 0 to $\sigma_{sat}$, where $\sigma_{i}\left(t\right)\;=\;0$ means that there is no pheromone at time step t, whereas $\sigma_{i}\left(t\right)\;=\sigma_{sat}$ means that the cell is saturated with pheromone at that time step. In a real-life AT study, pheromone concentration is measured in the number of molecules per cubic centimeter ($molecules/cm^{3}$). Whereas in ATM, pheromone concentration is measured in units of pheromone per cell (p units/cell).

It is important to note that [25] have another variable associated with the cells on the trail (resistance of the cell), which was used to analyze ATM on heterogeneous surfaces. However, as this paper’s focus is ant chemotaxis, we used a homogeneous trail throughout the simulations. Therefore, for mathematical simplicity, we avoided the variable in the present analysis.

In ATM, the ants are also indexed with a unique variable j (j=0,1,2,…,N), where N is the total number of ants in the simulation at the time of the measurements. As explained later in simulation scenarios, the number of ants changes over time. All ants in ATM have the following two variables assign to them.

$v_{j}\;\left(t\right)$ is the instantaneous velocity of ant j at time step t, measured in cells per time step (cells/time step). $v_{j}\;\left(t\right)$ is continuous and ranges from zero to one.$p_{j}\;\left(t\right)$ is the position of ant j on the trail at time step t and ranges from zero to L. Similar to $v_{j}\left(t\right),\;p_{j}\left(t\right)$ is also continuous.

For the simulations in this paper, it is assumed that: (i) ants cannot move backwards; (ii) the probability of forward motion is constant; and (iii) the distance travelled by an ant in a single time step depends on the cooperative perception through concentration of pheromone in the next cell. At every time step, the variables in the model simulation are updated in two stages.

### 2.1. Stage I: Ant Motion

The first stage of the update represents the ants’ behavior, which depends on a perception of a given ant of its surroundings (surroundings of an ant include the trail and other ants in the simulation). Here, based on the information about pheromones and the presence of ants in the cell ahead, the value of the instantaneous ant velocity ($v_{j}\;\left(t\right)$) is generated for a given time step t. At the end of Stage I, we obtain the value of $p_{j}\;\left(t+1\right)$ for each ant and scan the values of $s_{i}\left(t+1\right)$ for each cell. The ants’ positions and parameters are updated according to the following rules. If ant j is in cell i, the instantaneous velocity of that ant from cell i toward cell (i+1) depends on $s_{i+1}\left(t\right)$ and $\sigma_{i+1}\left(t\right)$, as shown in Equation (1).
(1)$$v_{j}\left(t\right)=\{\begin{array}{ccc} 0 & & ,\;if\;s_{i+1}=1\\ \{\begin{array}{c} max\left(v_{j}\left(t-1\right)-0.1,v_{min}\right)\\ v_{j}\left(t-1\right) \end{array} & with\;probability\;P\;\;\;\;\;\;\;\;\;\;with\;probability\;(1-P) & ,\;if\;s_{i+1}=0\;\&\;\sigma_{i+1}\left(t\right)<1\\ v_{min}+a\cdot \sigma_{i+1}\left(t\right) & & ,\;if\;s_{i+1}=0\;\&\;1\leq \sigma_{i+1}\left(t\right)<\sigma_{sat}\\ \left(v_{min}+a\cdot \sigma_{sat}\right) & & ,\;if\;s_{i+1}=0\;\&\;\sigma_{i+1}\left(t\right)\geq \;\sigma_{sat} \end{array}$$
(2)$$p_{j}\left(t+1\right)=p_{j}\left(t\right)+v_{j}\left(t\right)$$

For a given time step, there are four possible cases of the ant’s surroundings and corresponding ant’s behavior as follows:

The first case represents the action of an ant in cell i, if another ant occupies the next cell. In that case, the former ant cannot move forward, which is represented by $v_{j}\left(t\right)\;=\;0$ (study in [21] have indicated an absence of overtaking in AT). It is important to note that the first case represents exclusion dynamics, which ATM has inherited from Total Asymmetric Simple Exclusion Process (TASEP) model. As explained in [25,29], the TASEP model is the foundation model for ATM. The exclusion dynamics explain above plays an important role in the later analysis.

In the second case, the next cell contains no ant $\left(s_{i+1}\left(t\right)\;=\;0\right)$, but the pheromone concentration is also meager. Ants are sensitive to the pheromone concentration, where ants can detect the pheromone even at an extremely low concentration [18]. Thus, we assumed that ants could detect pheromones for $\sigma_{i+1}\left(t\right)\;>\;0$. However, for the ants to differentiate pheromone concentrations, the concentration needs to be above a certain level [18,19]. Thus, we also assume that, although ants can detect pheromone for $\sigma_{i+1}\left(t\right)\leq \;1$, ants cannot differentiate in the concentration if $\sigma_{i+1}\left(t\right)\leq \;1$. In such a scenario, although the ants might perceive the trail, the meager pheromone concentration will limit ants’ capacity to differentiate the concentration and perceive traffic scenario. Thus, considering the unknowns, to avoid wastage of energy and collision due to a high velocity, we assumed that the ants reduce their velocity or move with $v_{min}$, whichever is higher, and the probability P gives the probability of this event (change in velocity). Conversely, the ants maintain the same velocity as t−1 with probability (1 − P). In the second case, we assumed P = 0.7 (>0.5) representing the ants’ higher sensitivity to a meager pheromone concentration. Similar to [25,29], to have a larger velocity range for simulated ants, we also chose $v_{min}\;=\;0.15\;cells/time\;steps$.

In the third case, the next cell contains no ants $\left(s_{i+1}\left(t\right)\;=\;0\right)$, but it contains a differentiable level of pheromones below saturation $\left(1<\sigma_{i+1}\left(t\right)<\sigma_{sat}\right)$. In this case, the ant can perceive its surrounding through differentiation of pheromone concentration. Hence, the ant’s instantaneous velocity depends on the pheromone concentration in the next cell. We use a deterministic equation to represent velocity changes related to the third case. Deterministic dynamics represent ants’ high sensitivity to pheromones and low inertia on the trail. In this scenario, the value of the prefactor ${}^{\prime }a{}^{\prime }$ is decided based on $\sigma_{sat}$ and $v_{max}$ (upper-velocity limit (=1)), where for the functioning of the model within given velocity limits, ($\;a\cdot \sigma_{sat}\leq \;v_{max}\;-\;v_{min}$) needs to be satisfied. On the other hand, from [25,29], we know that a lower value of $a\cdot \sigma_{sat}$ will lead to a lower velocity range. Therefore, considering the above restriction, similar to [25,29], we selected $a\cdot \sigma_{sat}\;=\;0.8$. Whereas considering that in the range of $5\;<\sigma_{sat}\leq \;80$, ATM simulation remains independent of $\sigma_{sat}$, for mathematical simplicity, similar to [25,29], we chose $\sigma_{sat}\;=\;80\;(p\;units/cell)$, and a = 0.01. Although here in the model definition, we arbitrarily defined $\sigma_{sat}\;=\;80(p\;units/cell)$, which is the extreme value in the above mentioned $\sigma_{sat}$ range, the later sections of the paper show that the above assumption of $\sigma_{sat}$ does not affects the ATM simulation which correspond to real life AT. Whereas the same assumption allows us to analyse different chemotaxis parameters independent of $\sigma_{sat}$, which is important for our study.

The last case represents a scenario where the next cell contains no ant. At the same time, it represents a scenario where the pheromone level in the cell is above saturation. In such a scenario, the velocity of the given ant becomes equal to the velocity at saturation $\left(\sigma_{sat}\right)$, which we calculated by using the third case equation $\left(\;v_{min}\;+\;a\cdot \sigma_{sat}\right)$.

As specified by Equation (2), the new position $p_{j}\left(t+1\right)$ of the given ant is calculated by adding the position of the ant at time t to the distance traveled in unit time ($v_{j}\left(t\right)$).

### 2.2. Stage II: Pheromone Updating

In the second stage of the update, changes in surrounding and their effects on pheromone are presented for a given time step. For every time step, the concentration of pheromone on the trail changes for two reasons: (i) evaporation due to environmental factors (usually, the effect of surrounding on the evaporation rate $\left(r_{e}\right)$ remains constant if the surroundings remain approximately unchanged) and (ii) pheromone accumulation due to further discharge of the pheromone by the ants (in one-time step, an ant releases an amount τ (p units/time step) of pheromone, referred to as a pheromone unit). At the end of Stage II, we obtain the subset $\sigma_{i}\left(t\;+\;1\right)$ using the subsets $s_{i}\;\left(t\right)$ and $\sigma_{i}\left(t\right)$, and the velocity of the ant ($ant_{j}$) in the cell ($cell_{i}$) as follows: Evaporation:(3)$$\sigma_{i}^{\prime }\left(t+1\right)=\sigma_{i}\left(t\right)-\left(\sigma_{i}\left(t\right)\cdot r_{e}\right),\;if\;\sigma_{i}\left(t\right)>0$$Accumulation: (4)$$\sigma_{i}\left(t+1\right)=\{\begin{array}{c} \;\;\sigma_{i}^{\prime }\left(t+1\right)+\tau ,\;if\;s_{i}\left(t\right)=1\;\&\;\sigma_{i}^{\prime }\left(t+1\right)<\sigma_{sat}\;\&\;v_{j}\left(t\right)>0\\ \sigma_{sat},\;if\;s_{i}\left(t\right)=1\;\&\;\sigma_{\;i}^{\prime }\left(t+1\right)\geq \;\sigma_{sat} \end{array}$$

As given by Equation (3), evaporation depends on present pheromone concentration, where a certain fraction of the present pheromone evaporates from all cells at every time step depending on $r_{e}$. After evaporation, the remaining pheromone concentration on the cell is further affected by the addition of pheromone emitted by the ant in the cell at the same time step. As shown in Equation (4), for accumulation, there are two possibilities.

In the first case, the cell is occupied by an ant $\left(s_{i}\left(t\right)\;=\;1\right)$ that moved forward in the previous time step $\left(v_{j}\left(t\right)>0\right)$. Moreover, the pheromone concentration in the cell after evaporation ($\sigma_{i}^{\prime }\;\left(t+1\right)$) is below saturation. In this case, the ant releases a unit volume of pheromone in the cell, leading to increased pheromone concentration.

In the second case, an ant is occupying the cell $\left(s_{i}\left(t\right)\;=\;1\right)$, at the same time step, the pheromone in the cell is above saturation $\left(\sigma_{i}^{\prime }\;\left(t+1\right)\geq \;\sigma_{sat}\right)$. In the second case, the $\sigma_{i}\left(t+1\right)$ will be undifferentiable from $\sigma_{sat}$. Thus, it was assumed that the concentration remains at the saturation level $\left(\sigma_{sat}\right)$. It is important to acknowledge that assuming the pheromone detected and actual pheromone on the trail as the same could cause discrepancies between real-life AT and ATM simulation. However, as explained in the later analysis, the above assumption does not affect our analysis results and can be neglected. An overview of all the variables in ATM is provided in Table 1.

### 2.3. Simulation Scenarios

In this study, we wanted to analyze the AT on a trail that has formed over a period of time without any external interference. Thus, data were always collected after a considerable amount of time from the beginning of the simulation (collected data was from an established traffic flow). Moreover, the horizontal periodic boundary conditions were used to make the simulation scenario equivalent to a circular trail by connecting the last cell ($cell_{1000}$) to the first one ($cell_{1}$). Although natural ATs are open boundary systems, as we wanted AT simulations to form over a period of time without any external interference, we used the periodic boundary condition. Simultaneously, by using a trail of length 1000 cells, we ensured that the trail was long enough to make any ant’s self-interaction effects negligible, which arise under the periodic boundary condition with a short track [25,29].

At the start of the simulations, only one ant ($ant_{0}$) was introduced on the trail, and pheromone concentration in all cells was set at zero ($\sigma_{i}\left(0\right)=0$). Whereas during the simulation, at each time step, if $cell_{1}$ was empty, we introduced a new ant there ($cell_{1}$) with a probability known as the inflow rate. At the time of introduction, the new ant was assigned the variable j based on the N value before the introduction. The assigned j value was equal to the N. With the addition of new ants, the density in the simulation increases with time. In the present study, we conducted simulations until the density reached its limit (density = 1), where no further addition of ants was possible. Although foraging ants use multiple recruiting mechanisms, new foragers are usually recruited slowly [18]. Therefore, we were able to assume that the inflow of ants onto the trail was sufficiently low. For the simulations presented herein, we used an inflow rate of 0.001 (≪1), which allowed sufficient time for the traffic flow to become established in each density scenario.

## 3. Analysis of Pheromone Concentration and Its Implications for Cooperative Perception in the ATM

### 3.1. Evaporation Rate and Fundamental Diagrams

On a trail, ants release pheromone while moving forward, which other ants sense to understand trail and traffic conditions. Pheromone is a chemical substance, which ants use for communication [18,19]. If considered in isolation, pheromone released by an ant at a point on the trail contains information about the time passed since the ant passed that location. Considering a trail consisting of a single ant, the pheromone at any point on the trail contains information about the location and the time when the ant passed that location (location-time data). However, in real-life AT, the pheromone released by all ants have a similar chemical structure [18,19]. Hence, in AT’s CP&C, the pheromones released by all ants at any given location get added and cannot be used to understand the location-time information of an individual ant. Instead, as explained later, collectively location-time data (collected pheromone) at a given location from all the ants leads to an understanding of the traffic scenario through collective perception.

In the same CP&C, the evaporation represents a mechanism of discarding the collected pheromone (hence collected information) over time. For the CP&C of AT, mechanism for discarding the older information (pheromone evaporation) is a critical factor affecting cooperative perception in the system [11,13,15,25,26,28,30]. A trail with higher $r_{e}$ will discard the information quickly leading to the evaporation of the trail before it serves its purpose. Whereas a trail with low $r_{e}$ might also cause problems due to higher concentration and the need for higher cost for information storage and transmission [29,30]. Thus, for the CP&C of ants, it’s important to optimized the discarding mechanism by optimizing $r_{e}$. In this part of the paper, we present an analysis of different $r_{e}$ values and their effect on the CP&C in AT using ATM simulations. As shown in Figure 2, the fundamental diagrams of the relationships (a) the average velocity–density and (b) the flow–density, were obtained from ATM simulations for different $r_{e}$ values. For a time step in the simulations, we calculated (i) the average velocity ($v_{avg}$) by averaging the observed displacement of all of the agents for that time step, (ii) the density (d) by dividing  N at that time step with L (d = N/L), and (iii) the flow (f) by multiplying d by $v_{avg}$ at the same time step ($f\;=\;d\;\cdot \;v_{avg}$). The fundamental diagrams in Figure 2 are similar to the fundamental diagrams in [25]. However, as the purpose of the study in [25] was to present and validate ATM simulation, it did not provide a detailed analysis of $r_{e}$. In contrast, this paper focuses on the analysis of CP&C related parameters of AT. Thus here, we present a detailed analysis of $r_{e}$ and its effect on overall cooperative perception by using the fundamental diagrams and approximate mathematical analysis. In the case of ATM simulations, based on the fundamental diagrams in Figure 2, $r_{e}$ values in simulations can be divided into three ranges as follows.

#### 3.1.1. High-Medium Evaporation Rate ($0.5<r_{e}\leq 1)$

As Figure 2 shows, simulations with $0.5\;<\;r_{e}\leq \;1$ had monotonic $v_{avg}$ with respect to d. In high-medium $r_{e}$ range, $v_{avg}$ is seen to be approximatively constant for the free flow phase (FFP). For the jamming phase (JP), $v_{avg}$ has a monotonic decrease for corresponding d values. The flow-density graphs of ATM in high-medium $r_{e}$ range are similar to usual traffic systems [30]. In most traffic systems at lower density, a rising value of d leads to a linear increase in f values. Whereas after critical density, a rising d leads to a linear decrease in f. Interestingly, as discussed in [25,29,30], the ATM fundamental diagrams discussed above are similar to the fundamental diagrams for the TASEP model with a low hopping probability. From [25] we know that ATM used TASEP as a foundational model.

#### 3.1.2. Meager Evaporation Rate ($0\leq r_{e}\leq 0.001$)

In this range of $r_{e}$ value, a sharp rise in $v_{avg}$ is observed at a very low density (d < 0.1). Whereas thereafter, a gradual increase or constant $v_{avg}$ is observed for the rest of the FFP (see Figure 2); which leads to a peculiar flow–density graph that is distinct from most other traffic systems. In the JP of the fundamental diagrams, $v_{avg}$ is decreasing, which is similar to other traffic systems. As shown in Figure 2, the overall behavior of $v_{avg}$ in ATM in the above range of $r_{e}$ is non-monotonic, which causes anomalous fundamental diagrams. Although not precisely, the fundamental diagrams of ATM in the above range also resemble the fundamental diagrams of TASEP model but with high hopping probability [25].

#### 3.1.3. Low Evaporation Rate ($0.005<r_{e}<0.1)$

In the low $r_{e}$ range, as shown in Figure 2, initially constant $v_{avg}$ is observed for most of the FFP, whereas just before JP begins, a sharp rise in $v_{avg}$ (hereafter referred to as “rise-up”) is observed and then decreasing $v_{avg}$ is seen in JP. The above discussed non-monotonic behaviour of $v_{avg}$ in ATM leads to peculiar flow-density relation, where at the junction of FFP and JP, the flow in the system rise sharply before the system transit in JP. As explained in [25,30], the above behavior of rise-up in ATM is a hypothetical one. As ants in real-life AT avoid the JP, the rise-up between FFP and JP never happens naturally. Whereas, as explained in Appendix A, the rise-up in ATM happens because of the combination of two phenomena: (i) use of information about flow by ants to decide velocity and (ii) formation of the infinite cluster at high density. In vehicular traffic, as drivers primarily depend on the information about headway between themselves and the car ahead for velocity decision, vehicles’ velocity decreases monotonically with density. Thus rise-up never happens in vehicular traffic. Nevertheless, similar to meager $r_{e}$, low $r_{e}$ also leads to peculiar fundamental diagrams, which differ from those of typical traffic systems [11]. Furthermore, as shown in Figure 3, another exciting observation in low $r_{e}$ range is observed in regard to critical density, where initially decreasing $r_{e}$ lead to increasing critical density. Whereas later the same decreasing $r_{e}$ leads to a decrease in the critical density. In the above initial increase and later decrease in the critical density, the optimal critical density is observed around $r_{e}\approx \;0.02$, which was the value of $r_{e}$ used in [25] to represent real-life AT.

### 3.2. Pheromone Concentration and the Corresponding States of the CP&C in AT

Pheromone evaporation limits the pheromone concentration in a trail’s cells, affecting the cooperative perception in AT. Therefore, based on the above division of $r_{e}$ values, and corresponding pheromone concentration, we hypothesize the following three states of CP&C in ATM.

#### 3.2.1. Minimal Pheromone State

When the $r_{e}$ value for the ATM simulations is in the above-mentioned high-medium range, pheromone on the trail evaporates quickly. Thus, most of the time, ants in the simulations find only a meager pheromone concentration on the trail, which helps them identify the trail. However, it does not help them to understand traffic conditions in AT. In such a scenario, CP&C fail. The ants are forced to make their travel decisions based only on their immediate surroundings through limited visual and tacit information (based on the information about ant in the cell ahead). Whereas due to the lack of larger traffic information, ants cannot collaborate on any level with any other ants. The lack of collaboration considerably reduces AT efficiency, resulting in low $v_{avg}$ and low f in the simulations. In the paper, the state mentioned above (trail with high-medium $r_{e}$) is calleda “minimal pheromone trail”, and the corresponding CP&C state is called a “minimal state.” In the case of a high-medium $r_{e}$ value, owing to a meager pheromone concentration, most of the time ants travel with $v_{min}$ (constant) leading to the fundamental diagrams similar to the TASEP with low hopping probability [11].

#### 3.2.2. Inactive State

When the $r_{e}$ value for ATM simulation is in the above-explained meager range, the pheromone on the trail practically never evaporates. Whereas the boundary condition in the simulation leads to a traffic scenario where ants pass through the same point repeatedly. The combination of the above phenomena leads to a trail, where few ants can cause a rapid increase in pheromone, and hence a sharp rise in $v_{avg}$ and f, even at a low d value. In such a scenario, the pheromone concentration in all the cells will be the same, and it will depend on overall system parameters such as overall simulation density, overall simulation flow and others.

On the other hand, the same pheromone concentration becomes insensitive to the local environment of that cell. Furthermore, the above scenario makes pheromone concentration in a cell insensitive to the present time. Thus, in other words, the CP&C system in the above scenario will contain no information or only contains information about the overall system and time. The lack of local information makes the information useless for local decision. Whereas the overall system information is complex enough to cause information overload, making decentralized collaboration impossible [33]. Simultaneously, making the overall communication system inefficient. In the paper, the above communication state where although there is a large pheromone, it does not contain any substantial information is referred to as an “inactive state.” In an inactive state of CP&C, most of the ants experience near saturation pheromone. Therefore, travel with velocity near to $v_{max}$ (constant), which leads to fundamental diagrams similar to TASEP with a high hopping probability [11]. As [30] indicates, the inactive state represents a perfect system without errors. In a perfect, error-free system, motiles have unlimited energy and no delay. In such a system, irrespective of communication, motiles can move with maximum efficiency (instantaneous velocity) all the time and does not need cooperative perception to achieve efficiency. However, considering that usually the real life systems have delays and limited energy forcing them to move below maximum efficiency at some time or other, we suspect that the above system is a hypothetical one [25,29,30]. As shown in the later analysis, the above inactive state might only be observable in simulations, and real-life AT might never be entering in the above states.

#### 3.2.3. Active State

In the case of a low $r_{e}$ value, pheromone on the trail neither evaporates quickly nor gets saturated. In such a scenario, the pheromone concentration in a cell will depend on the flow of ants through the cell in the recent past [29]. Thus, in the low $r_{e}$ range, CP&C system through the chemotaxis will convey information about flow in the recent past. The trail mentioned above (a trail created with low $r_{e}$ pheromone) is called as an “active trail” and the corresponding CP&C state an “active state.” As [25] indicates the active state AT leads to velocity management based on the pheromone concentration, which, as mentioned, provides information about the flow in the recent past. We also know from [25] that active trail AT leads to the creation of multiple platoons, and the platooning leads to emergent behavior of congestion-free AT.

As mentioned in [29], in the active state, the pheromone concentration ahead of an ant depends on the flow through the cell in the recent past. However, if considered an ant from a platoon, then for that ant, the flow in the recent past will depend on the number of ants from the same platoon ahead of it. This happens because, as explained in [29], platoons in active trail ATM are separated by a considerable distance, and pheromone in a cell due to the platoon ahead evaporates to an indifferentiable level ($\sigma_{i}\leq \;1)$ before the following platoon reaches the cell. Thus, the pheromone concentration (hence the flow in the recent past) perceived by an ant could only be the result of the ants, who have already passed through the cell and who belong to the same platoon. In other words, the flow perceived by ants conveys the number of ants from the same platoon ahead of the ant.

On the other hand, in a traffic system, flow in the recent past also depends on different environmental and transportation factors in the same time period, including platoon density, velocity, and trail conditions. Thus, it could be argued that the pheromone concentration ahead of a given ant also conveys the environmental and transportation aspects discussed above. Therefore, based on the above discussion, we argue that for a given ant on an active trail in ATM, the instantaneous velocity is managed based on the perception of: (i) the number of ants from the same platoon ahead of the given ant; and (ii) the environmental and transportation factors. Moreover, we argue that the ants perceive the about information by understanding traffic flow in the recent past.

It is important to note that the information received by an ant in an active state is only about the limited surrounding of that ant. Ants receive limited information both from a space and a time perspective. From a space perspective, ants only receive information about the number of ants ahead in the same platoon. Whereas in the case of time, ants only receive information about the recent past. Thus, here onward, we define a concept of the extended local environment in space and time. The extended local environment from the space perspective includes the immediate local environment (cell ahead) and ants ahead from the same platoon. The extended environment from a time perspective includes the present time information and recent past information (the time when ants from the same platoon passed through the cell).

As shown in Figure 3, another observation of the fundamental diagrams in low $r_{e}$ range is also interesting one, where, as described above, initially the critical density is observed to increase with decreasing $r_{e}$ and then later the same is observed to decrease with decreasing $r_{e}$. As described above, in the above initial increase and latter decrease, the critical density is optimised at $r_{e}\approx \;0.02$, which was the $r_{e}$ value assuption in [25,29] to represent real-life AT. The above observation indicates that the real-life AT not only evolved to have a pheromone, which can function in an active state, but it also indicated that the AT is further biologically evolved to optimized the pheromone to have optimum free flow phase by having optimum $r_{e}$ ($r_{e}\approx \;0.02$) to have an optimum critical density (critical density ≈0.8).

### 3.3. Pheromone Concentration and Cooperative Perception in AT

The discussion presented in the minimal state’s and inactive state’s sections indicates ITS with two extreme and non-functional CP&C systems. In the minimal state, the information on the trail is discarded quickly, preventing CP&C from cooperative perception. Minimal state CP&C forces ants to make decisions based on the only immediate local environment. The immediate local environment of a given ant in ATM is the cell ahead of the ant, which the ants analyze using limited visual and tacit inputs. In the minimal state, the CP&C system is of minimal help for travel decision due to the lack of larger traffic scenario’s understanding. Although minimal state CP&C conveys the trail, it does not provide any substantial information for traffic management. The minimal state presents a communication system, where although ants need low resources (less pheromone (information) production, storage and less sensitive chemoreceptors (sensors)), the system is minimal in functionality due to the speed of information discarding. Thus create minimal value.

Whereas the inactive state represents another extreme non-functional CP&C system. As the information is practically never discarded in the inactive state, the CP&C is highly influenced by the distant past information, which makes the information insensitive to the local environment. Here the CP&C only provides information about the entire simulation system (ITS) and the entire past time when the trail was in use. In contrast, the same information becomes insensitive to the local environment and present time. In real-life ITS, due to the complex nature of interactions between motiles, self-organisation can only be done from bottom-up efforts [33,34]. In bottom-up self-organisation, constituent motiles (like vehicles or ants) collaborate on a manageable and local level, leading to emergent behaviors at the system level. Therefore, from individual motiles’ perspective, where information is needed to make decisions for self-optimization on the local level, the CP&C, like an inactive state, causes information overload to become useless. The information overloading happens because of the insensitivity of CP&C to the local level. In the inactive state, although there is a CP&C system that uses a lot of resources (higher pheromone (information) production, storage and highly functional chemoreceptors (sensors)), it does not lead to any meaningful insights. Perhaps that is why, as indicated in later parts of the paper, real-life AT never operates in the inactive state.

On the other hand, the discussion in the active state’s section presents a functional CP&C of AT. The discussion indicates that location-time data from individual ants (pheromone) in AT is collectively used to understand flow in the recent past. Whereas through the CP&C, individual ants use the flow information to understand traffic scenario. The traffic scenario here includes the information about platoon ahead, trail and transportation conditions in the recent past. Moreover, the analysis also indicates that the information discarding mechanism plays a crucial role in AT’s CP&C. The CP&C regularly discard past information to keep the information sensitive to the present time. The analysis indicates that the information perceived by ants in the active state is sensitive to the extended local environment both from the space and the time perspective. In the active state, the ants receive information about extended surrounding and past while ensuring that the information remains sensitive to the local environment and present time. Furthermore, the analysis indicates that ants are further evolved to optimize the critical density of AT by optimizing information discarding mechanism ($r_{e}\;$≈ 0.02).

Based on the analysis, we argue that an AT inspired CP&C for vehicular traffic can use individual vehicles’ location-time information to collectively understand traffic flow in the recent past. Using such information, vehicles can collaborate to achieve JAM similar to AT. The studies in [31,32], have already shown that JAM is implementable in vehicular traffic. Whereas the above analysis of ATM shows that flow information in the recent past can be used for the JAM. The analysis also showed that the AT inspired vehicular traffic’s CP&C needs optimization of its information discarding mechanism. The mechanism should be optimized to maintain the CP&C system in the active state where the information is sensitive to the recent past. Such CP&C will allow vehicles to achieve JAM on the local, manageable level (inside platoon). The active state system should facilitate synergy between individual vehicles’ needs and the entire ITS’s need by facilitating vehicles to make their local individual travel decisions while simultaneously allowing them to have manageable, bottom-up self-organisation.

## 4. Analysis of Pheromone Dynamics in Cooperative Perceptions of AT

Analysis and discussion in Section 3 elaborate on the general effect of the pheromone concentration on the CP&C system in AT. The analysis indicates that ants use flow information through chemotaxis for understanding the traffic scenario to achieve jam-free traffic. It also explains different aspects of the functional communication system in AT, including the sensitivity of the information to different traffic scenarios, and the relation between the sensitivity and resource management in AT. Moreover, the analysis explains and give us possible reasoning about findings from [25,29], which indicates that ants on a trail might have optimized trail pheromone evaporation rate ($r_{e}\;=\;0.02$) to function in an active trail state for foraging efficiency.

However, the results presented in Section 3 do not give us specific boundaries between different $r_{e}$ ranges. Similarly, it does not give us specific simulation values and definitions of the different pheromone concentration and the corresponding CP&C states. We also need further evidence to support our above findings and hypothesis in Section 3. Thus, this section presents a detailed analysis of pheromones’ impact on the CP&C system by introducing an approximate mathematical expression for pheromone-related scenarios. As explained in [30], the pheromone concentration on the trail has two dynamics that affect pheromones on a given cell: (i) pheromone aggregation due to passing of ant (aggregation of pheromone), (ii) pheromone depletion due to evaporation (depletion of pheromone). The following section begins with examining the effect of pheromone aggregation and the corresponding traffic scenario.

### 4.1. Aggregation of Pheromone

It is known that ants on a trail form multiple platoons, and ants use low $r_{e}$ value pheromone for trail creation [25,26,28,29]. Thus, for our analysis of pheromone aggregation, we consider a scenario in ATM simulation with low $r_{e}$ (Figure 4), where we assume an already established traffic flow. We also assume a scenario where ants have already created platoons.

At the start of the aggregation scenario (Figure 4a), the $cell_{X}$ (yellow cell) is considered, where a considerably long platoon is poised to pass through the cell. As the platoon is ready for passing, from [29,30], it is known that the leader of the platoon detect a pheromone concentration below the differentiable pheromone level (σ≤1), in the constant $v_{avg}$ phase (constant $v_{avg}$ phase is a phase in fundamental diagrams of ATM simulation with low $r_{e}$, where $v_{avg}$ remains constant irrespective of the density on the trail). Thus, based on the model description, the pheromone concentration in $cell_{X}$ at the beginning of the scenario (time = $t_{0}$) will be $\sigma_{X}\left(t_{0}\right)\;<\;1$. However, similar to [29,30], for simplicity, $\sigma_{X}\left(t_{0}\right)=0$ is considered. It was also assumed that ants in the platoon are closely spaced with no empty cell between them. Thus, during the passing of the platoon, the cell is always occupied. Assuming that it takes M time steps for the passing of the platoon, where M is a considerably high number of steps, at the end state ($t_{M}$), simulation reach the scenario explained in Figure 4b, where the pheromone in cell at $t_{M}$ is given by [29,30],
(5)$$\sigma_{X}\left(t_{M}\right)=\tau \cdot \;{\left(1-r_{e}\right)}^{M}+\tau \cdot {\left(1-r_{e}\right)}^{M-1}+\ldots +\tau \cdot {\left(1-r_{e}\right)}^{1}$$
(6)$$\sigma_{X}\left(t_{M}\right)=\tau \cdot \overset{M}{\underset{k=1}{\sum }}{\left(1-r_{e}\right)}^{k}$$

Mathematically, we know that Equation (6) converges for $\left(1-r_{e}\right)<1$. Therefore, rather than considering M time steps, we consider the extreme scenario with an infinite length platoon where the above expression converges.
(7)$$\sigma_{X}\left(t_{M}\right)=\tau \cdot \overset{\infty }{\underset{k=1}{\sum }}{\left(1-r_{e}\right)}^{k}$$
(8)$$\sigma_{X}\left(t_{M}\right)=\frac{\tau \cdot \left(1-r_{e}\right)}{r_{e}}$$

Combining the model dynamics with Equation (8), the pheromone in $cell_{X}$ at the end of the aggregation scenario can approximately be given as,
(9)$$\sigma_{X}\left(t_{M}\right)=\{\begin{array}{c} \frac{\tau \cdot \left(1-r_{e}\right)}{r_{e}},\;\;for\;\sigma_{x}\left(t_{M}\right)<\sigma_{sat}\\ \;\;\;\sigma_{sat},\;\;\;for\;\sigma_{x}\left(t_{M}\right)\geq \sigma_{sat} \end{array}$$

As mentioned above, and as can be inferred from Equation (9), the aggregated pheromone after passing of platoon through a given cell is converging with respect to the M value. In other words, Equation (9) indicates that the CP&C in aggregation scenario is highly sensitive to lower values of M and hence to the lower length of the platoon. Whereas as the M value increases the pheromone concentration become less and less sensitive to the value and hence platoon length. The above findings of Equation (9) support the hypothesis in Section 3, which argues that the CP&C system in AT is sensitive to the information in the recent past. Whereas the sensitivity decreases with time length and platoon length. Another interesting finding from the above analysis is observed in Figure 5, which presents $\sigma_{X}\left(t_{M}\right)$ for varying $r_{e}.$ As the figure shows the converging pheromone value for most of the active state $r_{e}$ simulations in below the $\sigma_{sat}$ value assumed in our analysis ($\sigma_{sat}\;=80\;p\;units/cell$). In other words, the observations showed that in the active state ATM simulation, pheromone concentration never reach the assumed $\sigma_{sat}$ value. This, thereby confirms our findings in [29,30]. The observation indicates that assumed $\sigma_{sat}$ in the modelling is misleading and is much higher than the $\sigma_{sat}$ representing real AT. The above analysis, along with previous findings in [29,30] indicates that considering pheromone chemoreceptors’ evolutionary optimization, the saturation pheromone concentration for AT might be dependent on $r_{e}$ and might be equal to the $\sigma_{X}\left(t_{M}\right)$ in the aggregation scenario. We argue that, as the analysis indicates that the $\sigma_{X}\left(t\right)$ never increase above the $\sigma_{X}\left(t_{M}\right)$ in aggregation scenario, the ants do not need an evolution of chemoreceptors to differentiate pheromone higher than the $\sigma_{X}\left(t_{M}\right)$. The analysis also confirms that the arbitrary value of $\sigma_{sat}$ in the model can be neglected [30]. Similarly, the analysis shows that the possible discrepancy that could have arrived due to assumption in accumulation dynamics in the model description, in practicality, does not affect the analysis (the assumption arguing that pheromone detected beyond saturation value and actual pheromone concentration as same). Along the same line, as shown in Figure 5 and calculated by Equation (9), for $r_{e}\;<\;0.012,\;\sigma_{x}\left(t_{M}\right)$ becomes saturated, which means that for the simulation scenario presented here, $r_{e}\;=\;0.012$ is the boundary between an active state and an inactive state. However, as discussed above $\sigma_{sat}$ for real-life AT might be dependent on $r_{e}$, where, with decreasing $r_{e}$, $\sigma_{sat}$ might be increased. Nevertheless, considering that the real-life platoons are finite, the discussion above also indicates that for a given set of $r_{e}$ and $\sigma_{sat}$ (assuming all other parameters unchanged), pheromone concentration on AT might not be exceeding that saturation value. Thus, we argue that perhaps real-life AT might never be entering the inactive state, and the observed inactive state in simulation might only be a hypothetical scenario, which does not represent any real-life AT. The above findings of ours are supported by the empirical observation in [21], which shows that density in real-life AT never exceeds the critical density (d<0.8) indicating the absence of the jamming phase and pheromone concentration that could saturate the chemotaxis system (inactive state).

### 4.2. Depletion of Pheromone

Taking the lead from Section 4.1, the current section presents an analysis for the depletion of the pheromone mentioned above (pheromone after aggregation $\left(\sigma_{x}\left(t_{M}\right)\;\mathrm{from}\;\mathrm{Equation}\;\left(9\right)\;\right)$ and its effect on cooperative perception in AT. Therefore, for the analysis here, $\sigma_{x}\left(t_{M}\right)$ becomes the $\sigma_{x}\left(t_{0}\right)$: starting time for the depletion scenario. As stated earlier, usually active trail ATM simulations have multiple platoons in constant $v_{avg}$ phase and the platoons are separated from each other by a substantial distance [25,29]. There are no ants or very few of them in the inter-platoon distance in an active state ATM. Thus, we can assume that once a platoon passes through the cell, the pheromone in that cell will deplete without further substantial addition. If the pheromone in the cell depletes for time D, then the relationship between $\sigma_{X}\left(t_{0}\right)$ and $\sigma_{X}\left(t_{D}\right)$ can be given as,
(10)$$\sigma_{X}\left(t_{D}\right)=\sigma_{X}\left(t_{0}\right)\cdot {\left(1-r_{e}\right)}^{D}$$

Equation (10) represents pheromone in $cell_{X}$ at the end of the depletion scenario (after D  time steps from the $\sigma_{X}\left(t_{0}\right)$ for depletion analysis). However, from [25,29], we know that in the constant $v_{avg}$ phase of simulation, pheromone accumulated due to the passing of the platoon ahead evaporates to an indifferentiable level before the following platoon reaches a given cell ($\sigma_{x}\left(t_{D}\right)\leq \;1$). Thus, considering D as the time headway between following platoon and the last ant from leading platoon and replacing $\sigma_{X}\left(t_{0}\right)$ by $\sigma_{X}\left(t_{M}\right)$, Equation (10) becomes,
(11)$$\sigma_{X}\left(t_{M}\right)\cdot {\left(1-r_{e}\right)}^{D}\leq 1$$

Using the logarithmic operator on both sides, we calculate the value of D as follows. D is the minimum time for which a leader of the following platoon will not reach the given cell from the beginning of the depletion scenario.
(12)$$D\;=\;-\frac{ln\left(\sigma_{x}\left(t_{M}\right)\right)}{ln\left(1-r_{e}\right)}$$

Based on Equation (12), Figure 6 presents the value of D  for various $r_{e}$, where, as shown in the figure, the D value exponentially decreases against increasing $r_{e}$. Based on the Figure 6, we can predict that for $r_{e}$ lower than a specific value, the D value will become so short that even the ants in the platoon will not experience a pheromone concentration from immediately preceding ants (concentration higher than the differentiable level). As Section 3.2.1 explains, the above mentioned high evaporation scenario is characterized by the minimal state. The mathematical expression also gives us the extended local environment radius that we discussed in Section 3.2.3 and also confirms the characteristics of the active states. In the case of high-medium $r_{e}$, the extended local environment is limited below the immediate local environment range (less than cell ahead), representing the minimal state. Whereas as the $r_{e}$ decrease, the extended local environment length increases both in time and space, which represents an expansion of extended local environment characterised by active state. The mathematical expression also shows that a further decrease in $r_{e}$ will lead to inactive trail by making the extended local environment equal to the overall system and insensitive to the local environment.

To find the boundary between the active state and the minimal state of ATM simulation, we consider the exclusion dynamics of ATM, which it has inherited from the TASEP model [25]. In the exclusion dynamics, the following agent does not move forward if the cell ahead is occupied by another agent (explained in case 1 from Equation (1)). As shown in Figure 7, the exclusion dynamics are expected often in a platoon where there will be many agents that have a distance headway ($headway\;=\;position_{leader}\;–\;Position_{follower}$) of greater than 1 (see Figure 7). We assume a minimal condition of the above scenario, where due to the exclusion dynamics, the ants have a distance of one cell between them. In the scenario in Figure 7, to have a minimal state, pheromone in the given cell must evaporate to an indifferentiable level before the following ant reaches the cell. Whereas, as the following ant is unable to differentiate the pheromone concentration and thereby travel with $v_{min}$, the approximate time taken by the following ant to reach the cell is given as in Equation (13).
(13)$$time\;=\frac{d}{v},\;\;where\;\;\;v\;=\;v_{min}\;\&\;\;d\;=\;1$$
(14)time =6.66 time steps

Using the values of Equations (4), (12) and (14), we calculate the $r_{e}$ value ($r_{e}\;=\;0.19$), which leads to the minimal state scenario for the simulations (Simulation scenario: $\sigma_{sat}\;=\;80\;p\;units/cell,\;v_{min}\;=\;0.15\;cell/time\;step,\;\tau \;=\;1\;p\;units$). Thus, the above analysis shows that in the presented ATM simulation, in the case of $r_{e}\;>\;0.19$, the CP&C will be in a minimal pheromone state. The analysis also verifies the characters of the different state of the trail, where it shows that in the minimal state, the information in the cell ahead erases before the next ant arrives in the cell. Whereas in the active state, the information in the cell is available and depends on the extended local environment.

Based on the above approximate mathematical analysis, different $r_{e}$ ranges and corresponding trail state for the ATM simulations in the simulation mentioned above, can be given as follows, Minimal state: $1\geq \;r_{e}\;>\;0.19$Active state: $0.19\geq \;r_{e}\geq \;0.012$Inactive state: $0.012\;>\;r_{e}\geq \;0$

The analysis presented above confirms the three communication states and the characteristics of those states in AT’ CP&C. The aggregation scenario analysis shows that the pheromone concentration ahead of a given ant provides the information about flow of ants through the cell in the recent past. The same analysis also confirms that the active state CP&C is sensitive to the information in the extended local environment both in space and time. Whereas from the perspective of the $r_{e}$ boundary between an inactive state and the active state, the analysis provides an expression to calculate the one. It also indicates that real-life AT might not be operating in the inactive state, and the observation in ATM simulation represents a hypothetical scenario. The same analysis also predicts some of the characteristics of ant chemoreceptor regarding saturation pheromone and its relationship with the evaporation rate in AT, which helped us clarify the ATM modelling discrepancies.

On the other hand, the depletion scenario analysis gives us the boundary between the active and minimal states. The analysis also confirms the characteristics of both the state. More importantly, the analysis also gives us the relation between the extended local environment and pheromone concentration (data) from both time and space dimensions. (In Appendix A, we have provided the verification of the above mathematical expressions.)

## 5. Analysis of Pheromone Emission Rate and Its Effect on CP&C System

From previous studies, we know that the Q:K ratio is a critical parameter for CP&C in the AT: (1) Q is the amount of pheromone (number of molecules) emitted by an ant in a one-time step, and (2) K is the smallest change in pheromone concentration (molecules per unit volume) at which the animal responds to the change [18,19]. In Section 4.1 of the paper (Equation (9), we note that Q represented by τ in the paper has an effect on pheromone concentration on the trail and hence CP&C in ATM simulation. Thus, to further understand the effects of τ on CP&C in ATM, we analyze the effect of different τ values on the traffic and information dynamics in ATM. Whereas, due to the limitations of the current model, the analysis of K is not considered in this paper. From a CP&C perspective, *τ* represents a weightage of information from a single ant in a one-time step. *Τ* has a similar effect as $r_{e}\;$ on CP&C, where ants with a lower *τ* value will create a pheromone trail that vanishes quickly, leading to a minimal pheromone trail. Whereas ants with a higher *τ* value will lead to slower depletion of pheromone, leading to ants faraway affecting traffic at any point on the trail, and perhaps causing inactive CP&C. Thus, ants need to optimize *τ* (weightage of information from single ant) to keep the CP&C in the active state.

To analyze the effects of τ on CP&C in AT, ATM was simulated with different τ in the active state, and the resulting fundamental diagrams are presented in Figure 8. As shown in the figure, with the rise in τ value, initially rise in FFP density was observed. However, after τ=1.5, stabilization occurred, with no further increase in FFP density.

From Equation (9), we note that pheromone aggregation after the platoon’s passing is directly proportional to the τ value in the simulation. Thus, with a rise in τ, pheromone aggregation increases, which consequently increases the efficiency of ants inside the platoon, leading to the higher velocity of ants in the platoon. In the case of two adjoining platoons, the above mentioned higher velocity due to higher τ leads to compact platooning, which consequently leads to higher inter-platoon headway for the following platoon. As explained in [25], an increase in the inter-platoon distance leads to a higher capability for jam absorption, leading to an increase in FFP density indicated in Figure 8.

However, the efficiency mentioned above can only be increased up to a certain limit, which is imposed by saturation concentration $\left(\sigma_{sat}\right)$. To understand these dynamics of τ with respect to $\sigma_{sat}$, $\sigma_{X}\left(t_{M}\right)\;$ from Equation (9) was analyzed for different τ values against different $r_{e}$ values. As shown in Figure 9, using Equation (9), we calculated pheromone aggregation with respect to different τ and $r_{e}$. For a given $r_{e}$, with an increase in τ, $\sigma_{X}\left(t_{M}\right)$ increases, but only up to a specific value. Any further increase in τ beyond the specific value leads to saturation of pheromone. For example, in the case of $r_{e}\;=\;0.02$, initially with an increase in τ, an increase in pheromone concentration is observed. However, τ = 1.63 (calculated from Equation (9)) leads to the saturation of pheromone, after which any rise in pheromone is undetected. This explains the stabilization in FFD density seen in Figure 9. However, as mentioned in the pheromone aggregation scenario analysis, the $\sigma_{sat}$ in real-life AT might be depending on convergent pheromone concentration in the aggregation scenario, and hence on $r_{e}$. By the same logic $\sigma_{sat}$ in real life, AT will also depend on τ value in the system, which means that in real life AT, hypothetically $\sigma_{sat}$ for the system should be increasing with a higher value of τ, leading to a further increase in FFP density with τ. Thus theoretically, the stabilization that we observed here might not be expected in real life AT and AT’s efficiency will continue to increase with τ.

Although the increasing τ is expected to increase the traffic system’s efficiency in AT, a larger value of τ needs higher production of pheromone and a larger pheromone storage, which is biologically expensive. A larger value of τ also means higher weightage for the information from a single ant, which will lead to the further extension of the extended local environment both in time and space. However, such a scenario will make the CP&C system comparatively less sensitive to a local environment in both time and space by increasing the importance of ants far away. It is also important to note that in the figure with increasing τ the range of $r_{e}$ for active state decrease. Considering that Ats are in an open environment, the decrease in the $r_{e}$ range makes CP&C vulnerable to fluctuations in surrounding environments. Thus, from the optimization perspective, it is preferable for ants to work with low τ. Our above finding argues that the information weightage for individual ant is optimized in trade-offs with the range of extended local environment and the system’s robustness. It also argues that the weightage of the information from single ant has a trade-off with storage and information transmission systems’ cost, which is crucial for any ITS. Interestingly, the above findings support and explain previous findings of ant physiology, which assert that, through biological evolution, ants have evolved to work with lower *τ* for efficiency reasons [18,19].

The discussion presented above suggests that in AT inspired CP&C for vehicular traffic, the weightage of data for an individual vehicle should be optimized for trade-offs with the range of extended local environment. The weightage of the information about an individual vehicle needs to be optimized to maintain the extended local environment to keep CP&C in the active state. Simultaneously, it suggests that the weightage of the information from motile will have trade-offs with information storage and transmission systems’ cost, which is crucial for any ITS. Suppose a higher weightage is given to information about an individual vehicle. In that case, the system will need to store that data longer, needing higher storage and transmission capabilities. Thus, it might be preferable to work with a lower weightage. The discussion also suggests that the lower weightage will make the system resilient to environmental changes, which might be a crucial advantage. However, care should be taken to ensure that weightage is high enough to avoid minimal state CP&C.

## 6. Discussion and Conclusions

The paper presents an investigation of the ATM to understand the CP&C system behind congestion-free AT. Using model simulations and simple mathematical representations of key scenarios in the model simulations, we analyzed various aspects of the CP&C system and their effects on AT. Our analysis gives detail insight into the communication and how it enables ants to collaborate for congestion-free transportation. The analysis also leads to some interesting observations and predictions about AT, which help us improve ATM modelling.

In CP&C of AT, ants collectively use chemotaxis to convey traffic scenario to individual ants. The pheromone from an ant on a trail contains location–time information of that ant throughout the transportation. Whereas in the same AT, pheromone from all the ants at any location gets accumulated. Individual ants use the accumulated pheromone to understand a larger traffic scenario. In AT’s CP&C, pheromone evaporation plays the role of information discarding mechanism, where older pheromone vanishes with time. In our analysis of pheromone concentration and its effects on cooperative perception in ATM, we found that different $r_{e}$ in the simulation lead to different pheromone concentration-dependent CP&C states: (1) the high-medium $r_{e}$ leads to a minimal state, (2) the meager $r_{e}$ leads to an inactive state, and (3) the low $r_{e}$ leads to an active state.

The high-medium $r_{e}$ represents CP&C, which discards the data quickly. In the high-medium $r_{e}$ due to quicker discarding, the information is reduced to minimal. Although the minimal information enables ants to identify the trail, it prevents ants from deducing any insight into the current traffic scenario. Thus, in this scenario, ants make their travel decisions only based on information about the immediate local environment. The lack of information in a minimal state prevents ants from establishing communication and understanding traffic scenario, which is vital for the collaboration. Thus, although the CP&C system needs fewer resources in the minimal state, the system only enables minimal communication and only achieves minimal synergy.

The meager $r_{e}$ represents CP&C, which practically never discard the data. In the meager $r_{e}$, due to the meager discarding of data, the data in the CP&C becomes saturated and insensitive to the local environment. Although the CP&C contains information about the entire system in the inactive state, the information’s insensitivity to the local environment makes it useless for the ants to make complex local decisions. Thus, even after using significant resources, the CP&C does not lead to any meaningful insight into the inactive state, causing wastage of resources. Perhaps that is why, as indicated in our analysis, real-life AT might be avoiding operations in the inactive state. The above two-state in CP&C represents two extreme and non-functional communication systems, which should be avoided.

On the other hand, the low $r_{e}$’s analysis showed that a pheromone trail resulting from low $r_{e}$ neither discard the information quickly nor does it get saturated. The same analysis indicates that in the low $r_{e}$, the pheromone concentration on the trail depends on the flow in the recent past. We refer to the CP&C state corresponds to the low $r_{e}$ as the active state. Previous studies have indicated that ATM simulations in the active state represent real-life AT.

In the active state CP&C, the ants’ location-time information is collectively converted into information about flow in the recent past. The ants use the flow information to understand traffic scenario and implement JAM. In the active state CP&C, the information discarding mechanism (pheromone evaporation) is optimized, where CP&C only provides information about flow in the recent past and discard the older data. The information that ants receive in the active state is sensitive to the local environment of the ant. Simultaneously, it also conveys information about the extended local environment both from space and time. Ants use the information to conduct JAM through collaboration in platoons (small groups). Because of the characters mentioned above, the active state CP&C system enables ants to establish synergy between individual decisions and the manageable bottom-up self-organisation leading to emergent behavior of jam-free traffic. The discarding mechanism in the CP&C system is not only optimized to function in the active state, but it is further optimized to have a free flow phase up to optimum density.

Moreover, to further investigate CP&C in AT, we analyzed pheromone dynamics and its effect on CP&C using approximate mathematical representations of critical simulated scenarios. Our mathematical analysis confirms the above mentioned three states and their characteristics. The characteristics of data collecting as well as discarding mechanism in the CP&C is confirmed. The expressions to define boundaries between different states are also provided. The analysis of pheromone aggregation also gives us some insight and predictions about ant chemoreceptor’s saturation limit. It predicts that the ant chemoreceptor’s saturation limit might be dependent on the pheromone evaporation rate and emission rate. The same analysis predicts that the inactive state observed in ATM simulation is just a hypothetical one that does not exist in real-life AT. Similarly, some of the ATM model’s discrepancies are clarified, which leads to model improvement. On the other hand, the depletion scenario analysis enabled us to understand the relation between the data discarding mechanism (evaporation rate) and the extended environment range.

Moving forward, the analysis of *τ* in ATM is presented, which gives us insight into the effect of changes in weightage of information from a single ant on the overall communication system. Our analysis indicates that theoretically, with an increase in weightage of information from a single ant, the traffic system’s efficiency increases. However, the analysis showed that the increase comes with higher cost and lower robustness of the CP&C system, indicating a trade-off between the weightage of individual motile’s data and the other two. The analysis also indicates that in the CP&C, the weightage of information for individual ant is optimized, with consideration for trade-offs with (i) the range of extended local environment, (ii) the system’s robustness, (iii) the cost of information storage and transmission. The weightage of information is optimized for maintaining the active state in the varying environment while avoiding the minimal state or the inactive state of CP&C. The analysis of *τ* in ATM also verified that ants prefer to work with a low *τ*, supporting previous physiological study about ants. The studies have suggested that, through biological evolution, ants have evolved to work with a low pheromone emission rate.

Most of the findings of this paper are new and thus need to be further verified and validated. Nevertheless, they give us a better understanding of the CP&C system that enables collaboration in the AT. The current findings of ants’ CP&C system and previous findings of JAM in AT could provide inspiration and insight for cooperative perception based ITS. Such an ITS could leverage AT’s learnings to utilize cooperative perception in collaboration for congestion-free transportation. Therefore, in future, further investigation needs to be done from that perspective. An investigation needs to be conducted to design and test AT inspired CP&C for JAM in vehicular traffic. Modified simulations of vehicular traffic and car-following models could be conducted considering AT inspired CP&C to investigate the efficiency of such a CP&C in vehicular traffic. A study comparing CP&C in the two systems should also be conducted to evaluate the two systems’ comparative efficiency. Simultaneously, the technical aspect for the design of such a CP&C needs to be looked at, which will include sensors designs.

## Figures and Tables

**Figure 1 sensors-21-02393-f001:**
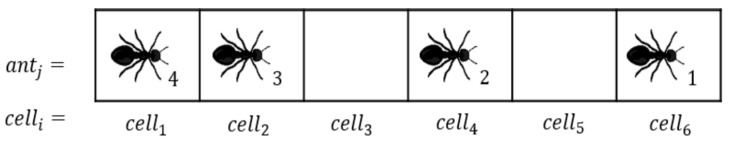
The single-lane ant-trail model (ATM) with a left to right motion is represented, where each cell of one-dimensional ant trail is indexed by i, and each ant is indexed by j. At any given time, there can be only one ant in any cell [25,29].

**Figure 2 sensors-21-02393-f002:**
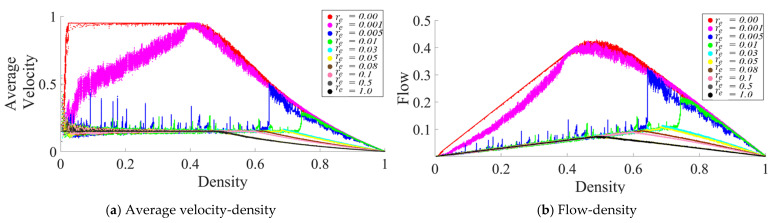
Fundamental diagrams for the ATM simulations with various $r_{e}$ values: (**a**) average velocity (cells/time step)− density relationship and (**b**) flow (ants/time step) − density relationship. Variables other than $r_{e}$ were kept constant: $L\;=\;1000\;cells,\;\sigma_{sat}\;=\;80\;p\;units/cell,\;v_{min}\;=\;0.15\;cells/time\;step,\;and\;\tau \;=\;1\;p\;units$ [25].

**Figure 3 sensors-21-02393-f003:**
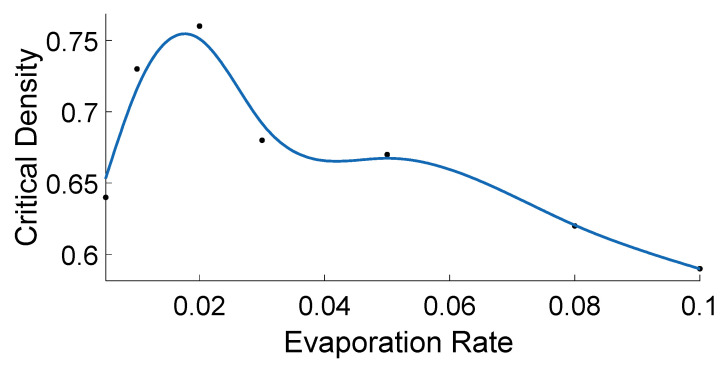
Critical densities of fundamental diagrams from ATM simulations are plotted against evaporation rate ($r_{e}$) of the corresponding simulations. For the simulations, variables other than $r_{e}$ were kept constant: $L\;=\;1000\;cells,\;\sigma_{sat}\;=\;80p\;units/cell,\;v_{min}\;=\;0.15\;cells/time\;step,and\;\tau \;=\;1\;p\;units$.

**Figure 4 sensors-21-02393-f004:**
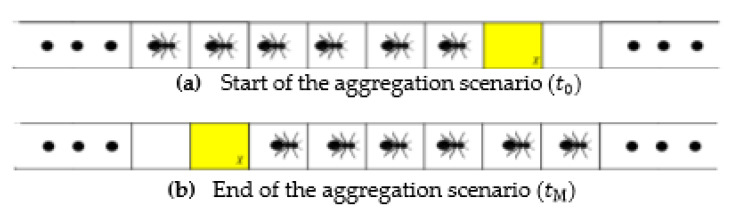
Schematic representation of (**a**) start of aggregation scenario ($time=t_{0}$), and (**b**) end of aggregation scenario ($time=t_{M}$). In this scenario, we consider the $cell_{X}$ (yellow cell), through which a considerable length of platoon will pass, leading to the aggregation of pheromone in that cell [29,30].

**Figure 5 sensors-21-02393-f005:**
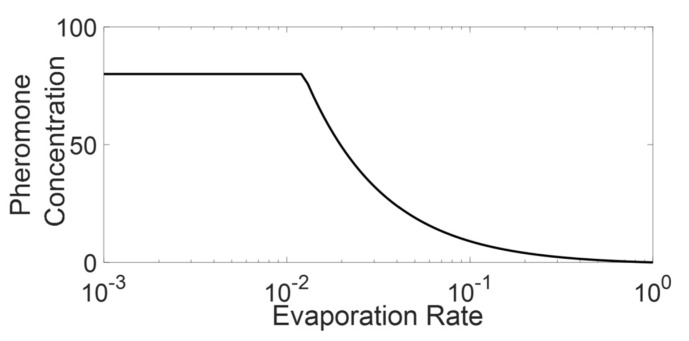
Approximate pheromone concentration at the end of the aggregation scenario ($t_{M}$) based on Equation (9) against evaporation rate is presented (for the calculations $\tau \;=\;1\;p\;units/time\;step,\sigma_{sat}\;=\;80\;p\;units/cell)$.

**Figure 6 sensors-21-02393-f006:**
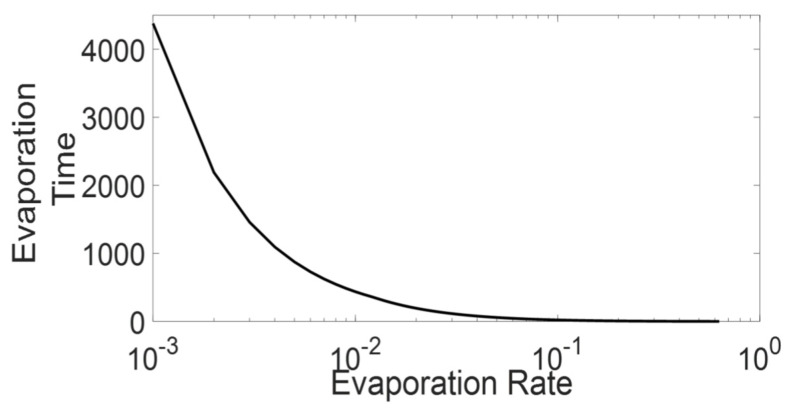
Approximate evaporation time required for the pheromone to decrease to the level below differentiable concentration ($\sigma_{x}\leq 1$) in depletion scenario is plotted against corresponding evaporation rate ($r_{e}$) (for the calculation $\tau \;=\;1\;p\;units\;/time\;step,\;\sigma_{sat}\;=\;80\;p\;unit\;/cell$).

**Figure 7 sensors-21-02393-f007:**
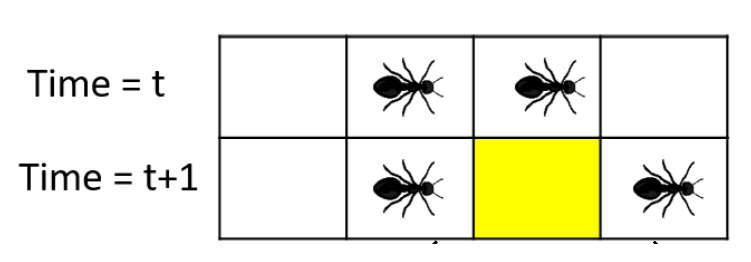
Schematic representation of the exclusion dynamics, which leads to creation of distance headway between adjoining ants from the same platoon in a single time step. In the diagram, the yellow cell represents the cell separating the rare end of the leading ant and the front end of the following ant.

**Figure 8 sensors-21-02393-f008:**
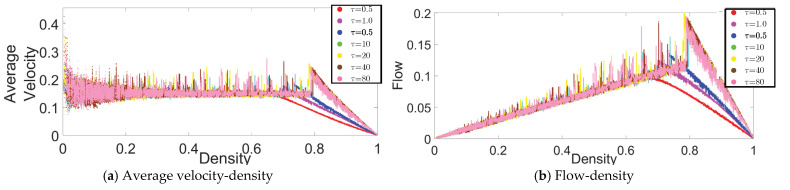
A fundamental diagrams from the ATM simulations with various τ values: (**a**) average velocity (cells/time step) − density relationship and (**b**) flow (ants/time step) − density relationship. Variables other than τ were kept constant: $L\;=\;1000\;cells,\;\sigma_{sat}\;=\;80\;p\;units\;/cell,\;v_{min}\;=\;0.15cells/time\;step,\;r_{e}=\;0.02$.

**Figure 9 sensors-21-02393-f009:**
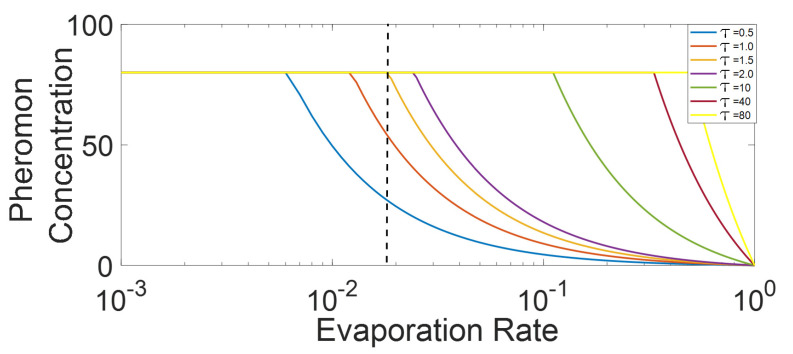
Approximate $\sigma_{sat}$ calculated for different τ is presented against evaporation rate ($r_{e}$). The dotted line represents the $r_{e}$ that has been used for the analysis in the Section 5 ($r_{e}\;=\;0.02,\sigma_{sat}\;=\;80\;p\;units\;/cell$).

**Table 1 sensors-21-02393-t001:** An overview of the parameters in the ant-trail model (ATM).

Description	Symbol
Unique identity of a cell in the trail	i
Presence or absence of an ant in the trail $cell_{i}$ at time *t*	$s_{i}\left(t\right)$
Pheromone concentration in the trail $cell_{i}$ at time *t*	$\sigma_{i}\left(t\right)\;$
Pheromone concentration saturation level	$\sigma_{sat}$
Unique identity of an ant in the simulation	j
Velocity of the $ant_{j}$ at time t	$v_{j}\;\left(t\right)\;$
Position of the $ant_{j}$ at time t	$p_{j}\;\left(t\right)$
The minimum velocity of an ant towards the cell with no pheromone and no other ant	$v_{min}$
Number of ants in simulation	N
Stochastic parameter in velocity reduction scenario	P
Trail length	L
Evaporation rate	$r_{e}$

## Data Availability

Not applicable.

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
