# Peer review of "Analysis of Cooperative Perception in Ant Traffic and Its Effects on Transportation System by Using a Congestion-Free Ant-Trail Model"

_sensors, 2021, doi:10.3390/s21072393_

Round 1
Reviewer 1 Report
The authors propose an evaluation method of an agent-based model for cooperative traffic simulation based on ants' behaviour. This paper doesn't present a new model, but further analyses of an existing ant-trail model. Although the paper is well written and structured, it needs major improvements before being considered for publication.
Detailed comments:
- Although it is clear that this paper represents an extension of previous works by the same authors, I suggest specifying better the contribution of this work according to previous existing models.
- Since the usefulness of the model should be extended to vehicular traffic considering the effect of cooperative intelligent transportation systems, it is not clear how this model could be applied as a road network model and how the model variables (e.g. pheromone, evaporation rate, etc.) correspond to cooperative road traffic variables. Please explain throughout the whole paper.
- To evaluate the effectiveness in simulating road traffic, I think that a comparison with other state-of-the-art cellular automata models should be considered.
Minor comments:
- Please improve eq. 1 for better readability.
- Please report all variables measure units (e.g., line 199 and all graphs).
- Please proofread the paper since some typos occur (e.g., "M time step" in line 520; "given as follow" in line 614, etc.).
Reviewer 2 Report
This paper conducts a comprehensive study on ant transportation. Extensive experimental results on various situations are analyzed. My suggestions for this paper are given below:
1) This is a pure simulation study. It is not clear whether the results can accord with real-world ant transportation systems. The authors did not use real-world ant data to guide their experiments, neither demonstrate that their results are consistent with real-world applications.
2) The authors stated that the knowledge of ant transportation can be used to guide ITSs. However, the similarities and differences between vehicle and ant traffic are not well-illustrated, e.g., it is hard to explain why there are velocity increases in the critical point in Figure 9. There is no similar phenomenon in vehicle traffic. Please give more demonstration on how to use this simulation system to study real-world human traffic problems.
Round 2
Reviewer 1 Report
The authors addressed all previous comments properly.
In my opinion, the paper is now suitable for publication in the journal.